# Tag Tree-Guided Multi-grained Alignment for Multi-Domain Short Video Recommendation

## ABSTRACT

Multi-Domain Rcommendation (MDR) aims to leverage data from multiple domains to enhance recommendations through overlapping users or items. However, extreme overlap sparsity in some applications makes it challenging for existing multi-domain models to capture domain-shared information. Moreover, the sparse overlapping users or items result in a cold start problem in every single domain and hinder feature space alignment of different domains, posing a challenge for joint optimization across domains. However, in multi-domain short video recommendation, we identify two key characteristics that can greatly alleviate the overlapping sparsity issue and enable domain alignment. (1) The following relations between users and publishers exhibit strong preferences and a concentration effect, as popular video publishers, who constitute a small portion of all users, are followed by a majority of users across various domains. (2) The tag tree structure shared by all videos can help facilitate multi-grained alignment across multiple domains. Based on these characteristics, we propose tag tree-guided multi-grained alignment with publisher enhancement for multi-domain video recommendation. Our model integrates publisher and tag nodes into the user-video bipartite graph as central nodes, enabling user and video alignment across all domains via graph propagation. Then, we propose a tag tree-guided decomposition method to obtain hierarchical graphs for multi-grained alignment. Further, we design tree-guided contrastive learning methods to capture the intra-level and inter-level node relations respectively. Finally, extensive experiments on two real-world short video recommendation datasets demonstrate the effectiveness of our model.

## CCS CONCEPTS

• **Information systems → Recommender systems**.

## KEYWORDS

Multi-domain recommendation, Multi-grained domain alignment, Tree structure, Popular publisher enhancement

**ACM Reference Format:**
. 2024. Tag Tree-Guided Multi-grained Alignment for Multi-Domain Short Video Recommendation. In *ACM MULTIMEDIA 2024, Melbourne, Australia*. ACM, New York, NY, USA, 9 pages. https://doi.org/xxxxx

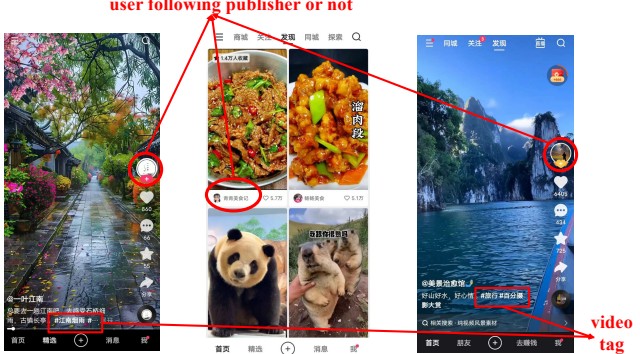

user following publisher or not

video tag

(a) Featured-Video (b) Double-Columned Discovery (c) Single-Columned Swift Slide

**Figure 1: Multiple scenarios in a popular short video platform. Only part of the tag information is displayed.**

## 1 INTRODUCTION

Aiming to provide users with multimodal content that caters to their diverse interests, short video recommendation plays a pivotal role in internet entertainment. With the advance of deep learning techniques, many deep learning models [2, 3, 5, 6] have been widely deployed in short video recommendation services, significantly enhancing user experience by accurately predicting preferences for unviewed short videos.

In response to users' varied needs and commercial objectives, short video platforms, such as Tiktok[1] and Kuaishou[2], generally offer multiple recommendation scenarios. For example, a popular short video platform has scenarios such as *Featured-Video*, *Double-Columned Discovery*, and *Single-Columned Swift Slide* scenarios, as depicted in Figure 1. These scenarios are similar to the multiple scenarios described in research [1] and can be treated as different specific recommendation domains of users and short videos. Generally, the overlapping users or short videos in these domains present the potential to enhance recommendations in each domain by leveraging users' interests in other domains. Nonetheless, in some applications, the overlap of users or items in different domains is scarce (i.e., *overlap sparsity* issue), which results in a specific domain being unable to fully exploit the additional abundant information from other domains. For instance, the analysis of 10 million instances of real interactions within the three aforementioned domains indicated the extreme overlap sparsity issue: **only 1.3% of users and 2.8% of short videos appeared simultaneously in these three domains**. Most existing multi-domain models [12, 13, 19] design a domain-shared module to model shared information across different domains. These models actually rely on overlapping users or items to learn shared information, making it difficult to adopt them for overlap sparsity issue. In addition, the overlap sparsity issue results in a cold start problem for a large

[1]https://www.tiktok.com/
[2]https://www.kuaishou.com/

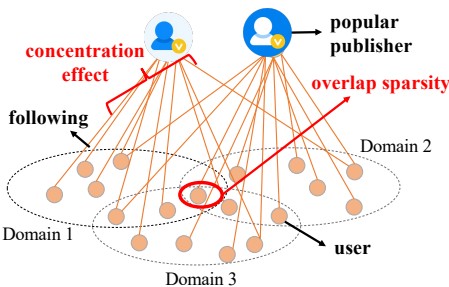

(a) Concentration effect of *following* relations

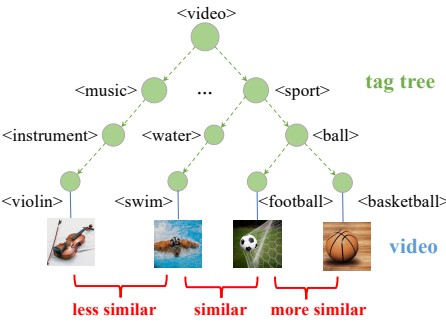

(b) Tag tree structure of short video recommendation

**Figure 2: Two key characteristics in multi-domain short video recommendation**

number of users in single domains, which limits the alignment of different feature spaces of different domains and poses a challenge for multi-domain joint optimization modeling.

However, we observe that the characteristics of two shared roles (i.e., user following publishers and video tags as shown in Figure 1) across different domains can greatly alleviate the overlap sparsity issue and achieve domain alignment. Specifically:

(1) Concentration effect of *following* relations: On short video platforms, users can assume dual roles as both viewers and publishers of videos. Intuitively, **following relations between viewers and publishers indicate stronger preferences** than the clicking relations. Thus, users who follow the same publisher in different domains are likely to share similar preferences. Moreover, **following relations exhibit a pronounced concentration effect** as Figure 2(a) depicted, where popular publishers accounting for a small number of all users connect to most users in multiple domains. This concentration effect, coupled with the strong preference of *following* relations, can help popular publishers serve as central nodes to align the preferences of non-overlapping users across different domains. Thereby, it can enhance preference knowledge sharing among domains. For simplicity, we refer to the popular publishers as publishers in the following.

(2) Tag tree-guided hierarchical similarity structure: Short videos across different domains share the same tag tree structure. As shown in Figure 2(b), videos related to <violin>, <football>, <swim> and <basketball> all share a common root tag, indicating their coarse-grained similarity. However, the degree of similarity among these videos varies at different levels of tag granularity. Thus, **the tag**

tree structure can serve as the supervision information to capture the hierarchical similarity of videos, enabling multi-grained alignment of non-overlapping videos across different domains. Additionally, video publishers typically exhibit a stable tag-releasing style. That is, although a single publisher may release a great many videos, the tags of these videos are generally similar or the same. Thereby, we can introduce the *releasing* relations to utilize the tag tree structure to model the similarity of publisher styles across domains. In addition to the above-mentioned *following* relation, it can further enhance the similarity of user preferences and alleviate the overlap sparsity issue.

Drawing on the two characteristics, we propose **T**ag tree-**G**uided **M**ulti-grained **A**lignment with **P**ublisher **E**nhancement (TGMAPE) for multi-domain short video recommendation. Specifically, we introduce publisher and tag nodes, along with their associated relations, into the bipartite graph of users and videos to construct a tag and publisher enhanced heterogeneous graph. In this graph, the publisher and tag nodes serve as central bridge nodes to enable user and video alignment across all domains. Then, we propose a tag tree-guided decomposition method, which decomposes the graph into hierarchical graphs for multi-grained alignment. Furthermore, to further address the sparsity issue, we design two contrastive learning functions to model the tree-guided intra-level and inter-level node relations respectively. These contrastive learning functions enhance the modeling of relations between node representations by reinforcing the similarities and differences among nodes within the tree structure. Finally, extensive experiments on two real-world short video recommendation datasets demonstrate the effectiveness of our model.

## 2 RELATED WORK

### 2.1 Single-Domain Recommendation

Existing recommendation models mainly focus on single-domain modeling. Traditional models, such as Collaborative Filtering (CF), assume users exhibiting similar behavior patterns share similar preferences [2, 6, 9]. Subsequent research [11, 20, 29, 32] pays attention to decoding users' evolving preferences from their historical behaviors. For instance, DIN [32] employs attention mechanisms to capture target item-related interests from user behaviors. Naturally, the interactions between users and items in recommendation systems form a graph structure. As Graph Neural Networks (GNNs) [27] have demonstrated the remarkable ability to process graph-structured data, many researchers have explored a great variety of GNN methods [8, 23, 25] to enhance recommendation performance. For instance, Graph Convolutional Network (GCN) [10] has been employed to factorize user-item rating matrices into user and item embedding matrices for recommendation. LightGCN[8] removes nonlinear activation and feature transformation in GCN. Nowadays, GNNs have served as important components in various recommendation models [22, 26, 30] to encode complex relations. In this work, we design tag tree-guided hierarchical multi-graphs for multi-grained alignment across different domains. Different from traditional recommendation graphs, it involves tree-structured node-level relations across multi-graphs, which allows for more accurate representations of nodes within graphs.

## 2.2 Multi-Domain Recommendation

Multi-Domain Recommendation [31, 34] aims to develop a unified model from data across domains to serve users in all domains. Previous research draws inspiration from Multi-Task Learning (MTL) principles, such as MMoE [14] and PLE [21]. MMoE uses different gate networks for each domain to fuse shared domain experts while PLE extends MMoE by introducing domain-specific experts. Based on PLE, SAR-Net [18] adopts attention modules to capture users' across-domain interests. STAR [19] proposes a star-shaped network consisting of one centered network shared across all domains and the other domain-specific network tailored to each domain. To capture the difference of domains at a finer-grained level, AdaSparse [28] and PEPNet [1] learn domain-specific features at parameter level. In summary, most multi-domain recommendation models use both domain-shared and domain-specific structures to extract a user's shared and specific features across domains. However, when dealing with the overlap sparsity issue, it is challenging to learn non-overlapping users' specific and shared features from their interaction data in a single domain. Although EDDA [15] proposes to identify similar pairs for domain alignment, it still relies on sparse overlapping users or items. Different from existing models, our model introduces publishers and tag tree structure shared by multiple domains for multi-grained alignment.

## 3 PRELIMINARY

Multi-Domain Recommendation typically employs $\mathcal{D} = \{D_i\}_{i=1}^O$ to denote $O$ distinct domains. Each instance in $\mathcal{D}$ can be represented as $(\mathbf{x}, y)$, where $y \in \{0, 1\}$ is the binary label of $\mathbf{x}$. In the specific multi-domain short video recommendation, each instance $\mathbf{x}$ contains the user, video, publisher and tag entities. We use $\mathcal{U}$, $\mathcal{V}$, $\mathcal{P}$ to denote the universal sets of users, videos and publishers across all domains respectively. Additionally, we adopt $\mathcal{T} = \{\mathcal{T}^k\}_{k=0}^K$ to denote universal sets of tags in the tag tree, where $\mathcal{T}^k$ represents the tag set at $k$-th level and $K$ indicates the depth of the tree. In short video recommendation, the *effective view* label, where if the viewing time exceeds 50% of the video's total duration, and 0 otherwise, is more significant than other interaction labels, such as clicks. Thus, we adopt the *effective view* label as $y$ in our work.

Each positive interaction instance $\mathbf{x} = (u, v, P, T, d_i)$ (i.e., $(\mathbf{x}, y = 1) \in D_i$) represents that in the $i$-th domain $d_i$, user $u$ following publisher list $P = [p_1, p_2, \cdots, p_M]$ effectively views video $v$ corresponding to tag list $T = [t^0, t^1, \cdots, t^K]$, where $M$ denotes the number of publishers that $u$ follows and $t^k \in \mathcal{T}^k$.

Moreover, publishers typically exhibit stability in the tag information of their released videos. The stable tag information can help model the multi-grained similarity between different publishers, which in turn facilitates the alignment of users across various domains. Thus, we introduce the *releasing* relation set $\mathcal{E}_{PT}$, where $\{(p, t'^k)\}_{k=0}^K \subset \mathcal{E}_{PT}$ indicates that publisher $p$ frequently releases videos tagged with $[t'^0, t'^1, \cdots, t'^K]$, where $t'^k \in \mathcal{T}^k$. Based on the above definitions, the multi-domain short video recommendation task can be defined as follows:

**Problem definition**: Given instance set $\mathcal{D}$ with *releasing* relation set $\mathcal{E}_{PT}$, this task should predict the label $y$ of each sample $\mathbf{x}$ in all short video recommendation domains accurately.

## 4 METHODOLOGY

In this section, we introduce TGMAPE for multi-domain recommendation, as depicted in Figure 3. In Section 4.1, we detail the construction of the tag and publisher-enhanced heterogeneous graph. Then, we introduce the tag tree-guided decomposition to generate hierarchical multi-graphs, enabling multi-grained alignment. Section 4.2 elaborates on the intra-level graph aggregation process based on the multi-graphs. Furthermore, in Section 4.3, we propose tree-based contrastive learning to capture both intra-level and inter-level node relationships. Lastly, Section 4.4 illustrates the inter-level fusion and prediction method.

### 4.1 Tag Tree-Guided Multi-Graphs Construction

*4.1.1 Tag and Publisher Enhanced Heterogeneous Graph Construction.* Different from existing multi-domain recommendation methods, our model exploits the tag tree structure for multi-grained alignment across different domains. The tree incorporates *parent-child relation* set $\mathcal{E}_{TT}$ among tags in $\mathcal{T}$. Specifically, if tag $t'$ is a parent tag of tag $t$, it can be denoted as $(t, t') \in \mathcal{E}_{TT}$. Thereby, our tag tree structure can be formulated as follows:

**Tag Tree Structure**. $Tree = \{\mathcal{T}^k\}_{k=0}^K \cup \mathcal{E}_{TT}$ satisfies:

- For each tag $t^{k+1} \in \mathcal{T}^{k+1}$, there exists a unique parent node $t^k \in \mathcal{T}^k$, i.e.,$(t^{k+1}, t^k) \in \mathcal{E}_{TT}$.
- For any two distinct levels $i, j$ within the tag tree, no tag node is shared, i.e., $\mathcal{T}^i \cap \mathcal{T}^j = \emptyset$.

Figure 2(b) illustrates a sample tag tree structure in which tag <instrument> is the parent tag of tag <violin>, for example. For each positive sample $\mathbf{x} = (u, v, P, T, d_i)$, multiple kinds of relations can be identified as follows:

- $\mathcal{E}_{UV}$: $(u, v) \in \mathcal{E}_{UV}$ represents that user $u \in \mathcal{U}$ interacts $v \in \mathcal{V}$ in domain $d_i$.
- $\mathcal{E}_{UP}$: $(u, p_i) \in \mathcal{E}_{UP}$ represents that user $u \in \mathcal{U}$ follows $p_i \in \mathcal{P}$. In $\mathcal{E}_{UP}$, **publishers serve as central nodes to connect most of users across different domains**, thus aligning sparse-overlapping users in different domains.
- $\mathcal{E}_{VT}$: $\{(v, t^k)\}_{k=0}^K \subset \mathcal{E}_{VT}$ represents that video $v \in \mathcal{V}$ corresponds to tag list $T = [t^0, t^1, \cdots, t^K]$ where $t^i$ represents the $i$-th level tag. For instance, as depicted in Figure 2(b), a violin-related video correspond to $T = [$<video>, <music>, <instrument>, <violin>$]$. In $\mathcal{E}_{VT}$, **tags serve as central nodes to connect all videos across different domains**. Besides, the hierarchical tree structure enables the multi-grained alignment.

Based on the above relation data, we can construct *Tag and Publisher Enhanced Heterogeneous Graph* as $\mathcal{G} = (\mathcal{U} \cup \mathcal{V} \cup \mathcal{P} \cup \mathcal{T}, \mathcal{E}_{UV} \cup \mathcal{E}_{UP} \cup \mathcal{E}_{VT} \cup \mathcal{E}_{PT} \cup \mathcal{E}_{TT})$, as depicted in Figure 3(a).

*4.1.2 Tag Tree-Guided Decomposition.* To facilitate the alignment of videos and users across multiple domains with different levels of granularity, we decompose the related relation sets $\mathcal{E}_{VT}$ and $\mathcal{E}_{PT}$ guided by the tag tree structure as follows:

$$\begin{aligned} \mathcal{E}_{VT}^{[k]} &= \{(v, t^k) \mid (v, t^k) \in \mathcal{E}_{VT}\}, \\ \mathcal{E}_{PT}^{[k]} &= \{(v, t^k) \mid (v, t^k) \in \mathcal{E}_{PT}\}. \end{aligned} \quad (1)$$

**Figure 3: The framework of TGMAPE**

Evidently, all tags of videos share a common root tag <video>. As the common tag does not facilitate differentiation among videos, we ignore the root level ($k = 0$) in the following. Accordingly, $\mathcal{G}$ is decomposed into the hierarchical multi-graphs: $\forall k = 1, 2, \cdots, K$,

$$\mathcal{G}^{[k]} = (\mathcal{U} \cup \mathcal{V} \cup \mathcal{P} \cup \mathcal{T}^k, \mathcal{E}_{UV} \cup \mathcal{E}_{UP} \cup \mathcal{E}_{VT}^{[k]} \cup \mathcal{E}_{PT}^{[k]}).$$

In particular, different from the simple superposition of multiple graphs, it incorporates a complex ***node-level relation*** set $\mathcal{E}_{TT}$ between hierarchical multiple graphs. Accordingly, the tag tree guided multi-graphs can be formulated as $\mathcal{G}^* = \{\mathcal{G}^{[k]}\}_{k=1}^{K} \cup \mathcal{E}_{TT}$, as depicted in Figure 3(b).

## 4.2 Intra-Level Graph Aggregation

Before aggregating graphs, we initialize the embeddings of users, videos, publishers, and tags. Aiming to achieve multi-grained alignment, we initialize $K$ group of embeddings for the $K$ hierarchical graphs accordingly. Given that the embedding and aggregation methods for the various entities within the multi-graphs are analogous, we choose to focus on the video entity for simplicity. Specifically, by transforming video $v \in \mathcal{V}$ into the $k$-th level video embedding matrix $E_v^{[k]} \in \mathbb{R}^{|\mathcal{V}| \times d}$ with $d$ denoting the embedding dimension, we obtain the initial embedding of video $v$ as $\mathbf{e}_v^{[k],0}$. Then the aggregation process can be formulated as follows:

$$\mathbf{e}_v^{[k],l+1} = \varphi(Agg(\{f^{uv}(\mathbf{e}_u^{[k],l}) \mid \forall u \in \mathcal{N}_v^{[k],U}\}),$$
$$Agg(\{f^{tv}(\mathbf{e}_t^{[k],l}) \mid \forall t \in \mathcal{N}_v^{[k],T}\})), \quad (2)$$

where $N_v^{[k],U}$ and $N_v^{[k],T}$ respectively denote the neighboring user and tag node sets of video $v$ in $\mathcal{G}^{[k]}$. In our work, we select the splitting function for $f(\cdot)$, mean pooling operation for $Agg(\cdot)$ and concatenation operation for $\varphi(\cdot)$ respectively, due to their efficiency and effectiveness through empirical analysis. Specifically, after splitting, the dimension of embedding $f(\mathbf{e})$ is reduced to $d/2$, such that $f^{uv}(\mathbf{e}_u^{[k],l}) \| f^{tv}((\mathbf{e}_t^{[k],l}) = \mathbf{e}_u^{[k],l}$ with $\cdot \| \cdot$ denoting the concatenation operation. Note that, in the edge set $\mathcal{E}_{UV} \cup \mathcal{E}_{UP} \cup \mathcal{E}_{VT}^{[k]} \cup \mathcal{E}_{PT}^{[k]}$ of graph $\mathcal{G}^{[k]}$, each node type is associated with two types of edges. For instance, regarding the user node type, it is associated with two distinct relation sets: $\mathcal{E}_{UP}$ and $\mathcal{E}_{UV}$. Upon setting the combination function $\phi$ as the concatenation operation, the dimension of embedding $\mathbf{e}_v^{[k],l+1}$ reverts to the original dimension $d$. **Through multi-layer graph propagation, a large number of non-overlapping users or videos can be aligned across different domains** with publisher or tag nodes as center bridge nodes.

Finally, the weighted-pooling operation is applied to aggregate representation by operating on the propagated $L$ layers as follows:

$$\mathbf{e}_v^{[k],*} = \sum_{l=0}^{L} \alpha_l \mathbf{e}_v^{[k],l}, \quad (3)$$

where $\alpha_l$ indicates the importance of the $l$-th layer representation in constituting the final representation. Following LightGCN [8], we set $\alpha_l$ as $\frac{1}{(l+1)}$ by default. Similarly, we obtain the aggregated representations as $\mathbf{e}_u^{[k],*}, \mathbf{e}_{t^k}^{[k],*}, \{\mathbf{e}_{p_m}^{[k],*}\}_{m=1}^{M}$ for user $u$, $k$th-level tag $t^k$ and $u$ following publisher list $P$ in $\mathcal{G}^{[k]}$.

**Video-aware publisher representation**. As users often have various interests, they may follow multiple publishers, such as game and music video publishers. To represent the user's following preferences for a particular video, we propose a **video-aware gate** mechanism that integrates information from the user's following publisher list. Specifically, the video-aware gating network $g$ produces a distribution over $M$ publishers based on the video-related input, and the final ensembled video-aware following preference representation $\mathbf{e}_P^{[k],*}$ can be formulated as:

$$
g(p_m) = \frac{\exp(\mathbf{e}_{p_m}^{[k],*} \times \mathbf{W}(\mathbf{e}_v^{[k],*} \| \mathbf{e}_{t^k}^{[k],*})^\top)}{\sum\limits_{j=1}^{M} \exp(\mathbf{e}_{p_j}^{[k],*} \times \mathbf{W}(\mathbf{e}_v^{[k],*} \| \mathbf{e}_{t^k}^{[k],*})^\top)},
$$

$$
\mathbf{e}_P^{[k],*} = \sum_{m=1}^{M} g(p_m) \mathbf{e}_{p_m}^{[k],*},
\tag{4}
$$

where $\mathbf{W} \in \mathbb{R}^{d \times 2d}$ serves to match the dimensions of vectors $\mathbf{e}_{p_j}^{[k],*}$ and $\mathbf{e}_v^{[k],*} \| \mathbf{e}_{t^k}^{[k],*}$; $\times$ indicates the matrix multiplication.

## 4.3 Tree-Based Contrastive Learning

In Section 4.2, we obtain the representations of tag-tree guided hierarchical graphs. To effectively capture node-level relations among these graphs, we design two types of constrastive learning. To illustrate them more clearly, we use Figure 3(c) as an example below.

(1) **inter-level contrastive learning**: Inter-level parent-child relations $\mathcal{E}_{TT}$ exist among different levels. For each node $A_1$ in $(k+1)$-th level, its parent node is $A$ rather than $B$ in $k$-th level. To capture the relation, we propose the inter-level parent-child indicator $\text{MLP}_e$ as supervision, where the score of positive inter-level node pairs $(A_1, A) \in \mathcal{E}_{TT}$ should be higher than negative pairs $(A_1, B) \notin \mathcal{E}_{TT}$. Naturally, contrastive learning is suitable for modeling the positive and negative pairs. We utilize the above tag information to enrich the video information. For a mini-batch of $N$ samples $\{\mathbf{x}_1, \mathbf{x}_2, \cdots, \mathbf{x}_N\}$, each sample contains a video $v_i$ and its corresponding tag list $[t_i^0, t_i^1, \cdots, t_i^K]$. For each sample $\mathbf{x}_i$, we sample a negative sample $\mathbf{x}_j$, such that for the $(k+1)$-th tag $t_i^{k+1}$, the $k$-th tag $t_j^k$ is not its parent tag $t_i^k$. Thereby, we collect paired contrastive training data $Y_e = \{(i,j)\}_{i=1}^N$, and formulate the inter-level contrastive loss accordingly:

$$
\mathcal{L}_e = -\frac{1}{K-1} \sum_{k=1}^{K-1} \sum_{(i,j) \in Y_e} C(\mathbf{e}_{v,i}^{[k+1]}, \mathbf{e}_{v,i}^{[k]}, \mathbf{e}_{v,j}^{[k]} | \text{MLP}_e),
$$
$$
C(\mathbf{a}, \mathbf{b}, \mathbf{c} | \phi) = \log(\sigma(\phi(\mathbf{a} \| \mathbf{b}) - \phi(\mathbf{a} \| \mathbf{c}))),
\tag{5}
$$

where $\sigma(\cdot)$ is the sigmoid activation.

(2) **intra-level contrastive learning**. Also, there exists a type of contrastive relation among nodes at the same level. To capture the relation, we propose the intra-level same-parent indicator $\text{MLP}_a$ as supervision. Specifically, the corresponding score of positive intra-level node pairs $(A_1, A_2)$ with the same parent node $A$ should be higher than negative pair $(A_1, B_1)$ with different parent nodes (i.e., $(A_1, A) \in \mathcal{E}_{TT}$, $(A_2, A) \in \mathcal{E}_{TT}$ and $(B_1, A) \notin \mathcal{E}_{TT}$). Thus, for each sample $\mathbf{x}_i$, we sample a positive sample $\mathbf{x}_j$ and a negative sample $\mathbf{x}_{j'}$, such that at the $(k+1)$-th level, $t_i^{k+1}$ and $t_j^{k+1}$ share the

same parent tag while $t_i^{k+1}$ and $t_{j'}^{k+1}$ do not. Thereby, we collect paired contrastive training data $Y_a = \{(i, j, j')\}_{i=1}^N$, and formulate the intra-level contrastive loss accordingly:

$$
\mathcal{L}_a = -\frac{1}{K-1} \sum_{k=1}^{K-1} \sum_{(i,j,j') \in Y_a} C(\mathbf{e}_{v,i}^{[k+1]}, \mathbf{e}_{v,j}^{[k+1]}, \mathbf{e}_{v,j'}^{[k+1]} | \text{MLP}_a),
\tag{6}
$$

where $C(\mathbf{a}, \mathbf{b}, \mathbf{c} | \phi)$ is the same as Eq. 5.

## 4.4 Inter-Level Fusion and Prediction

**Inter-Level Fusion**. After obtaining the hierarchical representations, we employ efficient mean pooling operation to fuse these representations. The inter-level video representation is then formulated as: $\mathbf{e}_v = \frac{1}{K} \sum_{k=1}^K \mathbf{e}_v^{[k],*}$. Similarly, the inter-level representations for users and publishers can be derived as $\mathbf{e}_u, \mathbf{e}_P$.

**Prediction and Optimization**. After obtaining the representations $\mathbf{e}_u, \mathbf{e}_v, \mathbf{e}_P$ and domain embedding $\mathbf{e}_{d_i}$, we concatenate them to obtain the overall representation of the instance $\mathbf{x} = (u, v, P, T, d_i)$ and then adopt the MLP to predict its label as follows:

$$
\hat{y} = \text{MLP}(\mathbf{e}_u \| \mathbf{e}_v \| \mathbf{e}_P \| \mathbf{e}_{d_i}),
\tag{7}
$$

where the last layer of MLP employs sigmod function as the activation function. Then, we employ the point-wise binary cross-entropy loss for a mini-batch of samples as follows:

$$
\mathcal{L}_o = -\sum_{i=1}^{O} \sum_{(\mathbf{x},y) \in D^i} y \log(\hat{y}) + (1-y) \log(1-\hat{y}).
\tag{8}
$$

Finally, the overall loss $\mathcal{L}$ is defined using hyper-parameters $\lambda_e$ and $\lambda_a$ as follows:

$$
\mathcal{L} = \mathcal{L}_o + \lambda_e \mathcal{L}_e + \lambda_a \mathcal{L}_a.
\tag{9}
$$

## 5 EXPERIMENTS

In this section, we present empirical results to demonstrate the effectiveness of our proposed TGMAPE. These experiments are designed to answer the following research questions: **RQ1** How does TGMAPE perform compared with state-of-the-art recommendation methods? **RQ2** What factors affect the performance of multi-domain recommendation? **RQ3** How do the hyper-parameters in TGMAPE impact multi-domain recommendation performance?

## 5.1 Experimental Settings

*5.1.1 Dataset Description.* The datasets utilized in this work were collected from a leading short video platform in China. The datasets were subsampled from multi-domain recommendation's logs, corresponding to two distinct time intervals[3]. These datasets, referred to as MDSVR-small and MDSVR-large, encompass data across three distinct domains: *Featured-Video*, *Double-Columned Discovery*, and *Single-Columned Swift Slide*, denoted as D1, D2, D3 respectively. Each dataset is split into training, validation, and test sets in a ratio of 8:1:1. Across both datasets, there are 5434 publishers that serve as central nodes. Besides, the tag tree had four levels (i.e., $K = 3$).

---

[3]We collected the data ourselves, due to the absence of public datasets that has analogous following relations and the tag tree structure. We plan to release part of our experimental datasets and code.

The number of tags at levels $k = 0, 1, 2, 3$ are $1, 39, 176, 414$ respectively. Table 1 provides a comprehensive overview of the datasets' characteristics, including the overlapping proportion of users or videos across the domains, as indicated in the *overlap* column. Notably, the overlap proportion for users and videos is relatively low, indicating that **the multi-domain datasets exhibit significant overlap sparsity**. In contrast, publisher and tag entities are shared across all domains.

**Table 1: The statistics of the two datasets.**

| Domain | D1 | D2 | D3 | All | Overlap |
|---|---|---|---|---|---|
| MDSVR-small | | | | | |
| #Samples | 545,162 | 44,859 | 886,765 | 1,476,786 | - |
| #User | 48,713 | 16,410 | 60,575 | 116,181 | 0.179% |
| #Video | 179,755 | 26,688 | 219,529 | 304,532 | 2.373% |
| MDSVR-large | | | | | |
| #Samples | 1,816,208 | 149,533 | 2,956,884 | 4,922,625 | - |
| #User | 64,344 | 28,938 | 74,091 | 146,067 | 0.583% |
| #Video | 318,711 | 63,772 | 368,977 | 589,392 | 4.668% |

*5.1.2 Evaluation Metrics.* Following many multi-domain recommendation works [18, 24], we choose two commonly-used metrics, **AUC** (Area Under the Curve) [4] and **RImp** (Relative Improvement) to evaluate the performance of the TGMAPE.

*5.1.3 Baselines.* To demonstrate the effectiveness of our proposed model, we compared it with three categories of existing recommendation models: Single Domain Recommendation (SDR) baselines, Multi-task Learning (MTL) models and MDR (Multi-domain Recommendation) models.

*(1) Single Domain Recommendation (SDR) Baselines.*

- **Single** integrates the embedding and prediction layers as detailed in Section 4.4.
- **NeuMF** [9] combines traditional matrix factorization with an MLP to simultaneously extract both low-dimensional and high-dimensional features.
- **NGCF** [23] obtains the representation of a node by aggregating its neighbor nodes and additionally uses element-wise multiplication to incorporate interactions.
- **LightGCN** [8] streamlines the GCN model by eliminating feature transformation and nonlinear activation layers, opting instead to directly apply neighborhood aggregation operations to update the embedded representation.
- **DCCF** [17] leverages an adaptive self-supervised augmentation to disentangle interests behind user-item interactions.

*(2) Multi-Task Learning (MTL) Baselines*

- **MMoE** [14] captures task relationships and shared representations through a set of different experts. We adapt MMoE for multi-domain recommendations by treating the task for each domain as an individual task.
- **PLE** [21] extends the MMoE by incorporating task-specific experts. Similarly, we apply PLE for multi-domain recommendations, building different experts for each domain.

*(3) Multi-Domain Recommendation (MDR) Baselines*

- **HMoE** [12] employs the implicit and explicit mix of expert structures to learn the relationships among domains.
- **STAR**[19] designs a domain-shared network and domain-specific networks to capture the shared and specific knowledge in each domain.
- **AdaSparse** [28] adopts neuron-level domain-aware weighting factors to measure the importance of neurons for different domains.
- **HiNet** [33] utilizes hierarchical information extraction and scenario-aware attentive network to cater to multi-task and multi-domain scenarios. For multi-domain recommendations, we adapt HiNet using a single task structure.
- **PEPNet** [1] incorporates dynamic parameters to more effectively capture domain-specific patterns.
- **EDDA** [15] disentangles knowledge across domains by separating model and embedding for inter-domain and intra-domain segments, and identifies similar user/item pairs from different domains through graph random walks.

**To ensure fair comparison, all baselines use the representations of $u, v, P, T, d^i$ in x as input.** Also, we introduce a modified version of our model, **TGMAPE(CPT)**, to validate that the performance enhancements of TGMAPE come from modeling the concentration effect of following relations and hierarchical multi-grained alignment with tags, rather than just adding tags and publisher information. Specifically, TGMAPE(CPT) aggregates user and video representations based on user-video bipartite graph and then embeds tag, publisher and domain representations as the input of MLP outlined in Section 4.4. For SDR methods, we train a separate model for each domain. For MTL methods, we treat the aggregated data from all domains as the input and recommendation for each domain as a single task, following previous MDR [7, 16, 19] setting.

*5.1.4 Implement Details.* All models in this study are implemented using PyTorch. In Section 5.3, we detail the impact of crucial hyper-parameters of tree-guided contrastive loss in our model, including the contrastive loss weights $\lambda_e$ and $\lambda_a$. Additionally, Section 5.4 examines the impact of other hyper-parameters, including the model depth $L$ and the dimension of latent embedding vectors $d$. We adopt the optimal configurations for these hyper-parameters. Under the above settings, all models are trained using Adam optimizer with a learning rate of 1e-4, batch size of 8196. The hidden sizes of MLP described in Section 4.4 are set as [64,32] by default. The hidden sizes of $MLP_e$ and $MLP_a$ for contrastive learning are set as [32]. To ensure fair comparison, we apply the above-mentioned settings across all models. Furthermore, we search for optimal values of the other hyper-parameters of the baseline models as suggested in their respective original papers. Ultimately, we employ the early stopping strategy based on the models' performance on the validation set to avoid overfitting.

## 5.2 Overall Performance Comparison (RQ1)

The experimental results for the two employed datasets across multiple domains are presented in Table 2, where we can observe that:

- TGMAPE consistently outperforms all baselines across all datasets and domains, **demonstrating the effectiveness**

**Table 2: Performance Comparison: The overall performance over the two datasets. "*"denotes that the best-performing method significantly outperforms the second-best one on the paired t-test (p-value < 0.05).**

| Model | MDSVR-small | | | | | | MDSVR-large | | | | | |
|---|---|---|---|---|---|---|---|---|---|---|---|---|
| | D1 | | D2 | | D3 | | D1 | | D2 | | D3 | |
| | AUC | RImp | AUC | RImp | AUC | RImp | AUC | RImp | AUC | RImp | AUC | RImp |
| Single | 0.7348 | - | 0.6662 | - | 0.7450 | - | 0.7554 | - | 0.7142 | - | 0.7666 | - |
| NeuMF | 0.7347 | -0.01% | 0.6673 | +0.17% | 0.7456 | +0.08% | 0.7528 | -0.34% | 0.7162 | +0.28% | 0.7674 | +0.10% |
| NGCF | 0.7354 | +0.08% | 0.6680 | +0.27% | 0.7458 | +0.11% | 0.7565 | +0.15% | 0.7106 | -0.50% | 0.7689 | +0.30% |
| LightGCN | 0.7360 | +0.16% | 0.6684 | +0.33% | 0.7469 | +0.26% | 0.7605 | +0.68% | 0.7162 | +0.28% | 0.7694 | +0.37% |
| DCCF | 0.7381 | +0.45% | 0.6553 | -1.64% | 0.7450 | 0.00% | 0.7571 | +0.23% | 0.7033 | -1.53% | 0.7698 | +0.42% |
| MMoE | 0.7404 | +0.76% | 0.6524 | -2.07% | 0.7452 | +0.03% | 0.7651 | +1.28% | 0.7101 | -0.57% | 0.7713 | +0.61% |
| PLE | 0.7410 | +0.84% | 0.6483 | -2.69% | 0.7470 | +0.27% | 0.7653 | +1.31% | 0.7146 | +0.06% | 0.7716 | +0.65% |
| HMoE | 0.7412 | +0.87% | 0.6619 | -0.65% | 0.7464 | +0.19% | 0.7652 | +1.30% | 0.7043 | -1.39% | 0.7706 | +0.52% |
| STAR | 0.7413 | +0.88% | 0.6619 | -0.65% | 0.7452 | +0.03% | 0.7621 | +0.89% | 0.7073 | -0.97% | 0.7687 | +0.27% |
| AdaSparse | 0.7443 | +1.29% | 0.6639 | -0.35% | 0.7471 | +0.28% | 0.7641 | +1.15% | 0.7074 | -0.95% | 0.7710 | +0.57% |
| HiNet | 0.7417 | +0.94% | 0.6674 | +0.18% | 0.7447 | -0.04% | 0.7658 | +1.38% | 0.7163 | +0.29% | 0.7721 | +0.72% |
| PEPNet | 0.7420 | +0.98% | 0.6532 | -1.95% | _0.7492_ | +0.56% | _0.7664_ | +1.46% | 0.7009 | -1.86% | _0.7726_ | +0.78% |
| EDDA | 0.7402 | +0.73% | _0.6690_ | +0.42% | 0.7483 | +0.44% | 0.7639 | +1.13% | _0.7165_ | +0.32% | 0.7709 | +0.56% |
| TGMAPE(CPT) | _0.7447_ | +1.35% | 0.6585 | -1.16% | 0.7491 | +0.55% | 0.7652 | +1.30% | 0.7097 | -0.63% | 0.7714 | +0.63% |
| TGMAPE | **0.7488***  | +1.91% | **0.6733***  | +1.07% | **0.7545***  | +1.28% | **0.7695***  | +1.87% | **0.7204***  | +0.87% | **0.7764***  | +1.28% |

**of our proposed model in multi-domain short video recommendation**. Due to the large volume of data, *0.002 improvement in AUC is significant* [28].
- Compared with TGMAPE(CPT), which takes user following publishers and video corresponding tags as side information, our model gains 1.17% and 0.91% for MDSVR-small and MDSVR-large, respectively. This improvement can be attributed to our model's incorporation of publisher and tag nodes in graphs, which can act as central bridges to learn from different domains through graph propagation.
- Compared with the SDR methods, most MTL and MDR baselines perform better in D1 and D3 while performing worse in D2. This may be due to D2 having less data than the other domains, posing challenges for models to accurately learn the representations of non-overlapping users or videos in D2. In contrast, our model achieves the best performance across all three domains because it can significantly reduce the difficulty of modeling sparse domains in a domain alignment manner.

## 5.3 Ablation Study (RQ2)

*5.3.1 Impact of cores in multi-graphs.* We conducted five ablation versions to investigate the impact of the publisher concentration effect and tag tree-guided hierarchical alignment in our TGMAPE.

- TGMAPE (w/o publisher concentration) removes edges related to publishers to analyze the impact of publisher concentration effect.
- TGMAPE(w/o tag alignment) removes edges related to tags to examine the impact of hierarchical alignment.
- TGMAPE(coarse-grained alignment) retains only the $k = 1$ level of the graph to validate that single coarse-grained alignment is inferior to multi-grained alignment.

- TGMAPE(medium-grained alignment) retains only the $k = 2$ level of the graph to validate that single medium-grained alignment is inferior to multi-grained alignment.
- TGMAPE(fine-grained alignment) retains only the $k = 3$ level of the graph to validate that single fine-grained alignment is inferior to multi-grained alignment.

To ensure fairness, the prediction layers of these ablation baselines have the same structure as that of TGMAPE. Based on the results presented in Table 3, we can observe that:

- The enhanced performance of TGMAPE compared with TGMAPE(w/o publisher concentration) confirms the effectiveness of publisher concentration in our model.
- The significant improvement of TGMAPE compared with TGMAPE(w/o tag alignment) indicates the effectiveness of tag alignment for multi-domain video recommendation.
- In single-grained alignment baselines, increasing granularity does not necessarily improve model performance. Specifically, medium-grained alignment yields the best results in D1 and D3, while coarse-grained alignment performs best in D2. This may arise because videos categorized with finer tags may introduce more noise, negatively impacting accuracy. Conversely, a slightly coarser granularity can mitigate noise in tag categories while maintaining sufficient alignment information.
- The single-grained alignment models underperform compared to our multi-grained alignment model, TGMAPE. This is because **multi-grained alignment can capture node similarity across multiple domains at varying levels**.

*5.3.2 Impact of tree-guided contrastive learning.* Tree-guided contrastive learning methods are designed to capture node-level relations within hierarchical multi-graphs. We conduct the following experiments to analyze their impact step by step.

**Table 3: Ablation study. The best results are highlighted in boldface respectively.**

| Model | MDSVR-small | | | MDSVR-large | | |
|---|---|---|---|---|---|---|
| | D1 | D2 | D3 | D1 | D2 | D3 |
| TGMAPE(w/o publisher concentration) | 0.7463 | 0.6665 | 0.7519 | 0.7681 | 0.7131 | 0.7743 |
| TGMAPE(w/o tag alignment) | 0.7388 | 0.6559 | 0.7430 | 0.7628 | 0.7056 | 0.7694 |
| TGMAPE(coarse-grained alignment) | 0.7463 | 0.6699 | 0.7510 | 0.7663 | 0.7128 | 0.7723 |
| TGMAPE(medium-grained alignment) | 0.7471 | 0.6604 | 0.7520 | 0.7676 | 0.7109 | 0.7732 |
| TGMAPE(fine-grained alignment) | 0.7457 | 0.6572 | 0.7509 | 0.7664 | 0.7098 | 0.7723 |
| TGMAPE | **0.7488*** | **0.6733*** | **0.7545*** | **0.7695*** | **0.7204*** | **0.7764*** |

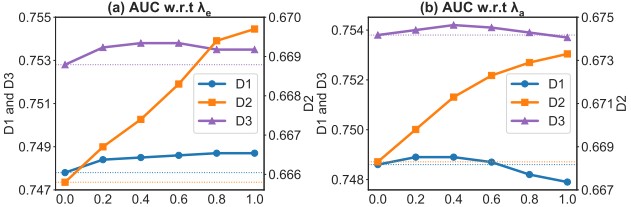

**Figure 4: Impact of tree-guided contrastive learning on MDSVR-small dataset.**

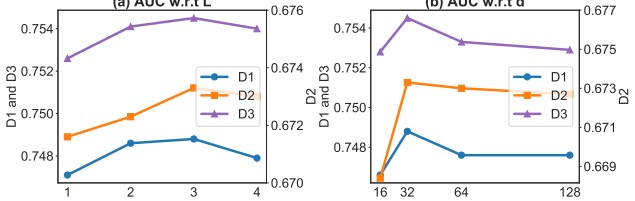

**Figure 5: Performance w.r.t aggregation depth $L$ and dimension $d$ on MDSVR-small dataset.**

*(1) Impact of inter-level relation contrastive learning.* We vary $\lambda_e$ within $\{0, 0.2, 0.4, 0.6, 0.8, 1.0\}$ with $\lambda_a = 0$. The results depicted in Figure 4(a) show that the model's performance initially improves and then declines as $\lambda_e$ increases. Notably, our model with inter-level relation contrastive learning (i.e., $\lambda_e \neq 0$) consistently outperforms the version without intra-level relation contrastive learning (i.e., $\lambda_e = 0$), **demonstrating inter-level contrastive learning can aid in tag tree guided multi-domain modeling**. Given that $\lambda_e = 0.8$ yields excellent AUC results for all domains, we adopt this value. After conducting a similar experiment, we set $\lambda_e = 0.4$ for MDSVR-large.

*(2) Impact of intra-level relation contrastive learning.* We vary $\lambda_a$ within $\{0, 0.2, 0.4, 0.6, 0.8, 1.0\}$ with $\lambda_e = 0.8$. The results in Figure 4(b) indicate that performance improves and then declines with increasing $\lambda_a$. Particularly, our model with $\lambda_a$ in $\{0.2, 0.4, 0.6\}$ outperforms the model without intra-level contrastive learning (i.e., $\lambda_a = 0$) for all domains, **demonstrating a proper loss weight of intra-level contrastive learning can aid in tag tree guided multi-domain modeling**. Given that $\lambda_a = 0.6$ yields excellent AUC results for all domains, we adopt this value. And set $\lambda_a = 0.2$ for MDSVR-large after a similar experiment.

In the case of D2 with the smallest sample sizes, the tag tree-guided contrast loss consistently enhances AUC across the tested range of loss weights. This may be because the tag tree-guided contrastive learning mechanism can leverage a substantial number of negative or positive video information sampled from other domains. By doing so, it allows for learning representations more accurately in the sparse domain D2.

### 5.4 Hyper-Parameter Studies (RQ3)

**Impact of Aggregation Depth**. Varying $L$ from 1 to 4, we observe that the performance initially improves and then declines, as

depicted in Figure 5(a). This trend can be attributed to a trade-off between the benefits of high-order aggregation information and the drawbacks of over-smoothing. For MDSVR-small, we set $L = 3$ as TGMAPE achieves the best performance at this value. Moreover, after conducting a similar experiment on MDSVR-large, we adopt the best-performing setting $L = 2$.

**Impact of Embedding Dimension**. Varying $d$ in $\{16, 32, 64, 128\}$, we find that TGMAPE's performance shows a trend of first increasing and then decreasing due to the overfitting issue, as depicted in Figure 5(b). Since TGMAPE performs best with the embedding dimension equaling 32, we set $d = 32$ for MDSVR-small dataset. After conducting similar experiments on the MDSVR-large dataset, we also adopt the best-performing setting $d = 32$.

### 6 CONCLUSION

In this work, we have identified two pivotal characteristics in multi-domain short video recommendation: users' concentrated following relations and videos' tag tree structure shared across domains, to greatly alleviate overlapping sparsity issue and facilitate domain alignment. Thereby, we propose a novel approach called tag tree-guided multi-grained alignment with publisher enhancement for multi-domain short video recommendation. Specifically, our model integrates publisher and tag nodes into the user-video bipartite graph, serving as central nodes to enable multi-domain alignment through graph propagation. Then, we devise a tag tree-guided decomposition method to generate hierarchical multi-graphs for multi-grained alignment. Further, we design contrastive learning methods to enhance the modeling of relations within the tree structure. Finally, extensive experiments confirm the effectiveness of our model.

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
