# OpenReview forum: "Tag Tree-Guided Multi-grained Alignment for Multi-Domain Short Video Recommendation"
_acmmm.org/ACMMM/2024/Conference — MM2024 Poster_

### Official Review · Reviewer_2Z5P · 2024-05-13

**Rating:** 3
**Confidence:** 3

**Summary:**

This paper focuses on multi-domain video recommendation. The author proposes Tag tree-guided Multi-grained Alignment with Publisher Enhancement (TGMAPE). It integrates two factors: preference relations between views and publishers and the hierarchical tag stage to jointly enhance the multi-domain performance across different domains.

**Strengths:**

S1: The structure of this paper is well-organized and easy to follow.

S2: This paper focuses on a real-world problem and concentrates on the problem of MDR video recommendation, which is rarely explored by researchers.

S3: The baseline comparison experiment is sufficient, including graph-based, multi-task, and multi-domain baselines.

**Limitations:**

L1: My primary concern with this paper is its limited scope. This method uses hierarchical video tree tags and the relationships between viewers and publishers, but for other datasets or platforms, this information might not be available. Moreover, this article may not serve as an effective reference for multi-domain research due to its limited scope.

L2: The description of domain splitting lacks clarity. What are the differences between Feature-video and single-columned swift slide? I could not tell the differences from Figure 1. The author should offer more comprehensive explanations of these domains.

L3: About the dataset and data process. The author's description of this part is too brief. Detailed information is required on the features used for user embeddings in section 4.2, what user features are used? Besides, how text-based video tag features are converted into tag nodes? Additionally, whether the labels refer to click logs or like logs and so on.

L4: As no experiments were conducted on public datasets, the author should provide dataset samples for reference and codes for validation.

**Suitability:**

2

---

### Official Review · Reviewer_WeV8 · 2024-05-19

**Rating:** 4
**Confidence:** 2

**Summary:**

This paper investigates the multi-domain recommendation problem. The problem itself is interesting in real practice. The authors propose Tag tree-Guided Multi-grained Alignment with Publisher Enhancement (TGMAPE) with tag tree-guided multi-graphs construction, intra-level graph aggregation and tree-based contrastive learning. The authors conduct extensive experiments to validate the proposed method which obtains SOTA performance.

**Strengths:**

S1: The paper is well-written and easy to follow.

S2: The methodology is technically sound.

S3: The authors conduct extensive experiments to validate the proposed method which obtains SOTA performance.

**Limitations:**

W1. Are the tree-like tag structures inherited by the datasets? More specifically, can the proposed method be used in the scenario when we only have tags (since it is easy to collect tags) in the recommendation datasets?

W2. [Minor] I encourage the authors to provide a pseudo algorithm table for the proposed method. Meanwhile, it is also important to discuss the time and space complexity of the proposed method.

**Suitability:**

3

---

### Official Review · Reviewer_ZvzJ · 2024-05-24

**Rating:** 4
**Confidence:** 3

**Summary:**

This paper tackles the overlap sparsity issue in multi-domain recommendation for short videos. The authors propose a tag tree-guided decomposition method to obtain hierarchical graphs for multi-grained alignment, and they also leverage tree-guided contrastive learning to further enhance representation learning. Experimental results validate its effectiveness on real-world datasets.

**Strengths:**

S1. This paper studies the practical and important problem of promoting Multi-Domain Short Video Recommendation by overcoming the overlap sparsity issue.

S2. The authors propose novel and interesting tag tree-guided hierarchical multi-graphs to align the interests of non-overlap users in different domains.

S3. The tree-based contrastive learning is solid and proved to be effective in enhancing representation learning.

S4. Generally, the paper is well-written and easy to follow.

**Limitations:**

W1. The promotion of the recommendation performance of the proposed model compared with the best SOTA is not significant.

W2. The notations in Section 4 METHODOLOGY are too complex and sometimes prevent readers from understanding the technology. It is suggested to simplify them or add a notation table for clarity.

W3. The authors did not open source the implementation code, so the reproducibility is poor.

**Suitability:**

3

---

### Official Review · Reviewer_nukU · 2024-05-26

**Rating:** 4
**Confidence:** 1

**Summary:**

The paper  presents a novel approach to enhance multi-domain recommendation (MDR) systems, specifically for short video platforms. The authors identify two key characteristics that can alleviate the overlapping sparsity issue and enable domain alignment in multi-domain short video recommendation: (1) the concentration effect of following relations between users and publishers, and (2) the shared tag tree structure among videos across domains. Based on these insights, the paper proposes a model named Tag Tree-Guided Multi-grained Alignment with Publisher Enhancement (TGMAPE) for multi-domain video recommendation.

**Strengths:**

1.Identifying key characteristics of multi-domain short video recommendation. Proposing a novel model that leverages these characteristics for improved recommendation. Introducing a tag tree-guided decomposition method and contrastive learning for better alignment and relation capturing.
2.The paper is well-written and presented in a clear and logical manner. The figures and diagrams effectively support the textual explanations, making the methodology and results easy to understand.
3.Conducting comprehensive experiments that validate the effectiveness of the proposed model.

**Limitations:**

It would be better to
1.Test the model's scalability to larger datasets or different types of domains.
2.Explore the impact of different hyperparameter settings on model performance.
3.Investigate the model's robustness to noise or changes in user preferences over time.

**Suitability:**

2

---

### Meta-Review · Area_Chair_ypxE · 2024-06-30

**Recommendation:** Accept (Poster)
**Confidence:** 5

**Metareview:**

Based on the reviewers' comments, this paper is well-written and demonstrates its novelty compared to the baselines, showing improvements. Therefore, I recommend that this paper be accepted by ACM MM 2024. Furthermore, I suggest the authors revise the paper according to the reviewers' comments to enhance its quality.